# Vitamin E and Metabolic Health: Relevance of Interactions with Other Micronutrients

**DOI:** 10.3390/antiox11091785

**Published:** 2022-09-09

**Authors:** Sijia Liao, Sylvia Oghogho Omage, Lisa Börmel, Stefan Kluge, Martin Schubert, Maria Wallert, Stefan Lorkowski

**Affiliations:** 1Institute of Nutritional Sciences, Friedrich Schiller University Jena, 07743 Jena, Germany; 2Competence Cluster for Nutrition and Cardiovascular Health (nutriCARD) Halle-Jena-Leipzig, 07743 Jena, Germany

**Keywords:** vitamin E, micronutrients, interactions, co-supplementation, metabolic syndrome, immune response

## Abstract

A hundred years have passed since vitamin E was identified as an essential micronutrient for mammals. Since then, many biological functions of vitamin E have been unraveled in both cell and animal models, including antioxidant and anti-inflammatory properties, as well as regulatory activities on cell signaling and gene expression. However, the bioavailability and physiological functions of vitamin E have been considerably shown to depend on lifestyle, genetic factors, and individual health conditions. Another important facet that has been considered less so far is the endogenous interaction with other nutrients. Accumulating evidence indicates that the interaction between vitamin E and other nutrients, especially those that are enriched by supplementation in humans, may explain at least some of the discrepancies observed in clinical trials. Meanwhile, increasing evidence suggests that the different forms of vitamin E metabolites and derivates also exhibit physiological activities, which are more potent and mediated via different pathways compared to the respective vitamin E precursors. In this review, possible molecular mechanisms between vitamin E and other nutritional factors are discussed and their potential impact on physiological and pathophysiological processes is evaluated using published co-supplementation studies.

## 1. The Immune System and Metabolic Disorders

With the rise of the severe acute respiratory syndrome coronavirus-2 (SARS-CoV-2), a worldwide pandemic started in December 2019. To date, the corona pandemic is still not under control even though a range of preventive measures including social distancing, quarantining, wearing mouth–nose masks and vaccines exist. The manifestation of this infection can be asymptomatic, with mild to moderate symptoms, or even severe symptoms which can be life-threatening. The infected patients, which progress into the severe stage, develop complications, such as pneumonia, multiple organ dysfunction, and septic shock [1]. Besides the development of effective treatments, challenges also include optimization of the immune response to reduce the risk of severe cases and even fatality caused by COVID-19 (coronavirus disease 2019). Several studies have demonstrated that poor metabolic health condition is associated with a higher risk of hospitalization as well as the severity of the disease [2,3,4]. People with metabolic diseases are more susceptible to severe symptoms and have higher risk of post-acute sequelae of COVID-19 [5]. Comorbidities, such as obesity, hypertension, diabetes, hypercholesterolemia, asthma, chronic obstructive pulmonary disease, chronic kidney disease, cardiovascular disease (CVD), osteoarthritis, and vitamin D deficiency, have been associated with severe COVID-19 [2]. For this reason, both prevention and treatment of the metabolic syndrome may reduce the risk of severe COVID-19 outcomes. 

It is evident that several micronutrients play crucial roles in the regulation of the immune system. The European Food Safety Authority (EFSA) has endorsed a list of micronutrients that contribute to maintaining the physiological function of the immune system including vitamin A, vitamin C, vitamin D, and the group of B vitamins, as well as trace elements such as zinc, selenium, copper, and iron [6]. In addition, some studies also suggested a meaningful role of vitamin E, vitamin K and magnesium against acute infectious respiratory diseases [7,8,9,10,11]. Meanwhile, dietary interventions have been considered as a key strategy to prevent and improve metabolic abnormalities in the long term.

Vitamin E is an essential nutrient that is known for its antioxidant and anti-inflammatory properties. Many observational, as well as interventional, studies have been aimed at elucidating the role of vitamin E in metabolic diseases. More intriguingly, there may exist interactions of vitamin E with the micronutrients vitamin A, C, and K, as well as selenium, magnesium, and zinc that could alter the biological activities of all involved micronutrients. Here, we summarize the scientific evidence for the underlying mechanisms by which vitamin E interacts with selected micronutrients and the effect of co-supplementation in different metabolic diseases. 

## 2. Characteristics of Vitamin E

Vitamin E is a fat-soluble organic micronutrient that helps to preserve human health. Vegetable oils, such as wheat germ, sunflower, corn germ, soybean, and rapeseed, are the primary dietary source of vitamin E for humans. It is also found in some nuts, fruits, and vegetables, such as almonds, avocados, spinach, and kale [12]. In 1922, vitamin E was first described as an essential dietary factor maintaining fertility in rats [13]. After that, numerous physiological activities of vitamin E have been discovered. Firstly, vitamin E serves as a free radical scavenger that protects biological membranes from lipid peroxidation. Beyond its classical function as an antioxidant, vitamin E is also involved in the modulation of enzyme activities and signaling cascades, as well as gene and protein expression (for details, see Section 2.3) [14,15,16]. Many cell responses including inflammatory responses, cell proliferation, programmed cell death, and lipid homeostasis have been shown to be affected by vitamin E. Despite the convincing findings from in vitro, as well as animal, studies, the conflicting results obtained from studies in humans failed to provide clear evidence for the benefits of vitamin E supplementation for disease prevention and therapy in patients with adequate levels of circulating vitamin E [17]. In summary, the cellular and molecular mechanisms by which vitamin E acts as well as its physiological relevance remains to be fully understood.

### 2.1. Different Forms of Vitamin E

Tocopherols (TOHs) and tocotrienols (T3s) are the two major groups of the naturally occurring molecules that are considered widely as vitamin E, and each of them has four analogues, namely α, β, γ, and δ. All forms of vitamin E are composed of a chromanol ring system and a hydrophobic side-chain, which is comprised of 16 carbon units. TOHs and T3s can be distinguished by the saturation of the side-chain, and the four analogues of each have different patterns of methylation of the chromanol ring. Due to different affinities to the hepatic transporter, the α-tocopherol transport protein (α-TTP), the eight forms of vitamin E reveal different biological activities in vivo. It is believed that binding to α-TTP prevents the molecule from degradation [18]. With the favor of preferentially binding to α-TTP, α-TOH is the form with the highest abundance in the human body and has been deemed as the most meaningful form of vitamin E [19]. It is currently being discussed whether α-TOH should be recognized as the only form of vitamin E due to its indispensable role in the prevention and treatment of vitamin E deficiency associated diseases such as familial isolated vitamin E deficiency (AVED) [20].

### 2.2. Bioavailability of Vitamin E

The bioavailability of vitamin E varies greatly between individuals and can be affected by several factors. The topic has been extensively reviewed in [21,22]; hence, it will be only briefly summarized here. Vitamin E in food, but especially in dietary supplements, is often present as an esterified molecule. After hydrolysis by pancreatic and intestinal digestive enzymes, vitamin E is released from food matrices and absorbed into micelles. Several proteins have been shown to affect the efficiency of vitamin E absorption in the intestine, namely the scavenger receptor class B type 1 (SR-BI), Niemann-Pick C1-like protein 1 (NPC1L1), and the ATP-binding cassette transporter of the sub-family A number 1 (ABCA1). The transport of vitamin E in blood is similar to that of cholesterol. Vitamin E in hepatocytes is transported and stabilized by α-TTP. As mentioned in Section 2.1., α-TOH is preferentially distributed to peripheral tissues due to its highest affinity to α-TTP. Thus, α-tocopheryl acetate, the stabilized form of α-TOH, and α-TOH are the most widely used forms of vitamin E in human trials. In addition to α-TTP, other cellular vitamin E binding proteins have been suggested in humans, such as supernatant protein factor (SPF) and tocopherol-associated protein (TAP). Possible factors that likely affect the bioavailability of vitamin E, i.e., its absorption and metabolism, such as the different forms of vitamin E itself, chemical modifications (e.g., esterification), the amount consumed at once, and interactions with other nutrients or drugs consumed in parallel, have been summarized by Borel et al. [22]. There are also host-related factors, such as vitamin E status, state of health, and genetic factors. In fact, lower absorption of α-TOH in patients with metabolic syndrome or the elderly has been reported [23,24]. It has been hypothesized that, probably due to increased inflammation and oxidative stress, the absorption in the small intestine as well as the hepatic trafficking of vitamin E are limited. These findings provide evidence that some populations may have a higher demand of vitamin E. From our point of view, blood concentrations of vitamin E should be measured before and after supplementation in study participants to properly evaluate its effect in human trials, since the bioavailability of vitamin E, and in turn the vitamin E status, is affected by numerous factors and individual conditions. At present, the blood concentrations and the concentration in organs of α-TOH as well as the level of the endogenous metabolite of α-TOH, namely α-carboxyethyl-hydroxychroman (CEHC), are used as biomarkers of vitamin E status [25]. 

### 2.3. Hepatic Metabolism of Vitamin E

As a lipophilic compound, circulation and distribution of vitamin E follow largely that of lipoprotein metabolism [21]. The enzymatic degradation of vitamin E occurs mostly in hepatic tissues, although its catabolism has been also described for the intestinal epithelium [21,26]. Vitamin E metabolism is extensively reviewed in [21,27]. It is assumed that the hepatic metabolism of TOHs is initiated with the enzymatic oxidation catalyzed by cytochrome P450 (CYP) 4F2 or CYP3A4 [28,29]. The resulting metabolites are the long-chain metabolites (LCM), namely, the initially formed 13’-tocopherol-hydroxychromanol (13’-OH) and the subsequently formed 13’-tocopherol-carboxychromanol (13’-COOH). Thereinto, 13’-COOH has emerged as a derivate of vitamin E with many regulatory functions, revealing superior and sometimes even different effects compared to its metabolic precursor [30,31,32,33,34,35,36]. The following β-oxidation shortens the side-chain of the LCM and results in the formation of several intermediate-chain metabolites (ICM) and, finally, the short-chain metabolite (SCM) of vitamin E, namely, CEHC or 3’-COOH. All the metabolites are excreted with feces or urine following conjugation with taurine, glycine, glucuronide, or sulfate [21]. It is evident that the T3s follow almost the same metabolic degradation route as the TOHs, except for one additional step that is required for removing the double bonds in the side-chain of the T3s, most likely via the same enzymes, namely, 2,4-dienoyl-CoA reductase and 3,2-enoyl-CoA isomerase, which are also involved in the degradation of unsaturated fatty acids [37].

### 2.4. Biological Activity and Physiological Relevance of Vitamin E

One of the most important functions of vitamin E is its antioxidant function, by which it protects polyunsaturated fatty acid (PUFA) in membranes, lipoproteins, and intracellular lipid droplets from oxidation. Free radicals attack the double bonds in PUFA of cellular phospholipids, causing irreversible damage to the cell membrane, eventually leading to cell death (Figure 1). Vitamin E, which is also found in membrane bilayers, quenches the peroxyl radicals by delivering the hydrogen atom of the hydroxyl group at the chromanol ring. The resulting tocopheroxyl/tocotrienoxyl radical can be restored to the original form in redox cycle reactions that cooperate with other endogenous antioxidants, including glutathione (GSH), ascorbic acid (vitamin C), and coenzyme Q (Figure 1). In vitro studies showed that the different forms of vitamin E exhibit various antioxidant capacities due to the relative H atom donating ability which depends on the position of methyl groups at the chromanol ring. Among them, α-TOH exhibits the highest ability to break the peroxidation chain [38]. For this reason, α-TOH, precisely the naturally occurring *RRR*-α-TOH, is regarded as the form of vitamin E with strongest antioxidant efficiency and has been therefore used most widely as a supplement when investigating the potential benefits of vitamin E in humans. Besides its role as a peroxyl radical scavenger, α-TOH is also found to enhance the antioxidant cascade of nuclear factor erythroid 2–related factor 2 (Nrf2) [39]. As a consequence, the expression of antioxidant enzymes such as catalase and superoxide dismutase (SOD) is induced to defend cells against intracellular oxidative stress.

Apart from its antioxidant function, several non-antioxidant properties of vitamin E have also been discovered in pre-clinical studies in the last few decades (reviewed in [14,40,41,42,43]). Vitamin E has been demonstrated to affect immune function, inflammatory response, lipid homeostasis, cell cycle, and apoptosis in different cell types. Particularly highlighted are the anti-inflammatory effects of vitamin E, with multiple underlying mechanisms, of which some have been also established in animal models. Amongst others the control of the activation of signaling pathways including mitogen-activated protein kinase (MAPK), nuclear factor kappa B (NF-κB), phosphatidylinositol-3-kinase/AKT kinase/mammalian target of rapamycin (PI3K/Akt/mTOR), Janus kinase (JAK)/signal transducers and activators of transcription (STAT) pathways, modulation of gene and protein expression of, for example, cyclooxygenase-2 (COX-2), microsomal prostaglandin E_2_ synthase-1 (mPGES-1), and inducible nitric oxide synthase (iNOS), regulation of enzyme activities of, for example, 5-lipoxygenase (LOX-5) were observed (intensively reviewed in [14,15,42]). 

Considering properties dependent and independent from its function as an antioxidant, vitamin E is believed to influence processes in various pathological conditions, such as obesity, diabetes, CVDs, and neurodegenerative diseases [40,43,44,45]. The occurrence of cardiovascular events is related to numerous factors such as endothelial dysfunction, enhanced oxidative stress, chronic inflammation, abnormal blood lipid level, and xenobiotic metabolism. Several in vitro studies revealed a positive role of vitamin E, especially α-TOH, in the maintenance of endothelial barrier functions, anti-inflammatory actions, prevention of oxidation of low-density lipoproteins (LDL) as well as regulation of lipid homeostasis [17]. The potential role of vitamin E supplementation in the prevention of metabolic disorders, such as CVD, diabetes, and fatty liver disease, has also been intensively studied in human trials (reviewed in [40,44,45]). Due to the essential role of vitamin E for the nervous system, the application of vitamin E against Alzheimer’s disease-associated pathology has also been investigated in human intervention trials (reviewed in [46]). Further, lower vitamin E status was associated with an increased risk of tumor development [47,48]. Due to these findings, the preventive role of vitamin E has been intensively discussed and also evaluated in human clinical trials [49]. On the one hand, data from observational studies confirm a strong association between higher vitamin E concentrations and lower risk of metabolic diseases [50]. On the other hand, the interventional studies varying strongly in study size, cohort selection, dosage, and duration of the treatment have presented controversial results, which make the role of vitamin E as therapeutic strategy more questionable. 

The officially recommended daily intake (RDI) of 12–15 mg/d vitamin E in Germany is based on its biological activity against lipid peroxidation [51]. Numerous foods including nuts, seeds, and some sorts of vegetable oil act as natural sources of vitamin E. Moreover, vitamin E is also widely used as a legal additive in food processing. Even so, it is frequently observed that vitamin E requirement cannot be completely covered by dietary intake [21]. According to the German Consumption Survey II (*Nationale Verzehrsstudie II*), conducted between 2005 to 2007, almost half of the investigated adults in Germany did not reach the RDI of vitamin E [52]. In addition to the intake via food, the intake via supplements should also be considered. The German Consumption Survey II also showed that with supplementation, only about 70% of the participants achieved the RDI of vitamin E [52]. Another study reported that more than half of the adults in the US used one or more dietary supplements daily [53]. Only 4% of the participants supplemented only vitamin E, but 32% of them took multivitamin/mineral supplements containing vitamin E to improve overall health [53]. In an assessment of several dietary surveys conducted in the USA, the UK, Germany, and the Netherlands, Troesch et al. [54] showed that more than 75% of the population in the UK and USA consume amounts of vitamin E below the country-specific RDI. In the Netherlands, 25–50% of women and 5–25% of men were below the RDI. This observation demonstrates the need to investigate how other nutrients may influence the potential beneficial effects of vitamin E. 

## 3. Nutritional Factors Interfering with the Effects of Vitamin E Supplementation

### 3.1. Vitamins

#### 3.1.1. Vitamin C

Vitamin C is a water-soluble antioxidant and an essential cofactor of enzymes for several biosynthetic pathways. It affects many aspects of the immune response (reviewed in [55]). It is required for proper barrier integrity and wound healing by regulation of collagen synthesis [55,56]. Vitamin C contributes to antioxidant regeneration and also regulates redox-sensitive signaling pathways, thereby reducing oxidative stress during the inflammatory response [55,56]. Vitamin C also affects the differentiation and proliferation of immune cells and participates in the modulation of plentiful immune activities [55]. Recently, there has been increased interest in the effect of high doses of vitamin C in critical illness, particularly in infectious diseases and the resulting systemic inflammatory responses [57]. The benefits of high-dose vitamin C intake has also been studied in patients with severe COVID-19 [58,59].

Under oxidative challenge, a network of antioxidants functions together to quench free radicals (Figure 1). For instance, the antioxidant properties of vitamins C and E can be continually restored by other redox systems, such as the thiol redox cycle [60]. As shown in Figure 1, after quenching lipid radicals, the oxidized form of vitamin E, i.e., the tocopheroxyl/tocotrienoxyl radical, reacts with ascorbic acid. The latter is subsequently oxidized while the oxidized form of vitamin E is restored to its native, i.e., reduced form. The oxidized form of vitamin C, i.e., the semi-ascorbyl radical or dehydroascorbate, can be regenerated enzymatically by thiol group containing antioxidants. This recycling is essential to maintain the intracellular concentration of the reduced form of vitamin E [61,62]. Evidence for the interaction between vitamins C and E is mostly obtained from in vitro studies and it is poorly understood how this interaction works in vivo. The interventional study of Hamilton et al. [63] analyzed the interaction of vitamin C and E and the effect of supplementation on their respective plasma concentrations. In an interventional study, 30 healthy adults received pharmacological doses of either vitamin C (500 mg/d) or E (73.5 mg/d *RRR*-α-TOH acetate). After six weeks, the concentration of both vitamins was determined. The study revealed an increased plasma concentrations of vitamin E after supplementation of vitamin C as well as increased plasma concentrations of vitamin C after supplementation of vitamin E. The activity of the antioxidant enzyme glutathione peroxidase (GPx) in plasma was enhanced, and concentrations of total cholesterol decreased at the same time in both groups. Jungert and colleagues also found a substantial positive interaction between plasma concentrations of vitamins C and E after tracing 399 subjects aged older than 60 years for a period of twelve years [64]. This interaction has been shown to be independent of body composition and lifestyle factors, including physical activity, smoking and dietary intake, but gets stronger with increasing age. 

In vitro studies have shown that vitamin C reverses the pro-oxidant form of vitamin E. Based on this concept, a combination of a supplementation with both vitamin C and E is expected to have beneficial effects against metabolic diseases. The result of the Alpha-Tocopherol-Beta-Carotene (ATBC) study indicated that vitamin E may cause harmful effects on health in a subgroup of male smokers [65]. Hemilä and co-workers [66,67] re-analyzed the data of the ATBC study with respect to a possible correlation of the dietary intake amounts of vitamin C and E and their influence on mortality. The analysis suggests that the effect of vitamin E can be beneficial but also harmful, depending on several factors, such as age, health condition, and personal habits, such as smoking. The reader is referred to the fact that Hemilä et al. [66,67] found that the dietary intake of vitamin C also alters the effect of vitamin E on mortality. Intake of the recommended dietary allowance of vitamin C, i.e., ≥90 mg/d, caused different effects of vitamin E in an age-dependent manner. In brief, in participants aged 50–62 years, the supplementation of vitamin E (50 mg/d α-TOH) increased mortality by 19%, whereas mortality was decreased by >40% in those aged 66–69 years. These findings demonstrate that the benefit of vitamin E supplementation in the elderly may be conditional on a high intake of vitamin C [66,67]. This finding may explain why high-dose vitamin E seem to increase all-cause mortality in some studies (for example, [68] or [69]). On the other hand, insufficient intake of vitamin C seems also to affect the beneficial effect of vitamin E. In circumstances that intake of vitamin C is below recommendations, supplementation of vitamin E failed to show benefits on mortality at any age [62,67]. Similar conclusions have been drawn by McKay and co-workers [70]. They followed 9704 healthy middle-aged men for ten years and found that higher plasma concentrations of both vitamin C and TOH were associated with lower risk of all-cause mortality and cardiovascular events [70]. 

Observational studies have consistently shown that the vitamins C and E in combination reduce the risk of metabolic disorders and CVD [64,66,71]. However, interventional studies revealed inconsistent results. A study investigated the effect of high-dose supplementation of vitamin E (537 mg/d α-TOH) and C (1000 mg/d) on oxidative stress in women with endometriosis and found that, after an intervention of eight-weeks, the biomarker of plasma oxidative stress, the malondialdehyde (MDA), and reactive oxygen species (ROS) levels were significantly decreased and the symptoms of endometriosis were reduced [72]. The reader should note that the investigation of the intake of vitamins E and C in reproductive-aged women revealed that 33.7% of the respondents did not reach the estimated average requirement of both vitamins with their diet [73]. Rafighi et al. [74] demonstrated that the supplementation of vitamin E (201 mg/d α-TOH) and C (266.7 mg/d) decreased blood glucose concentrations, oxidative stress and insulin resistance in 170 individuals with diabetes mellitus type 2 (T2DM). Several randomized controlled trials were performed investigating the impact of the co-supplementation of these vitamins on the improvement of hypertonic outcomes [75,76]. Positive results including reduced blood pressure and improved endothelial function have been reported after a supplementation of α-TOH for two to three months (for details, see Table 1). However, studies showing contradictory results have been reported. After a follow-up of eight years, the long-term supplementation of vitamin E (268 mg α-TOH per alternating days) and C (500 mg/d) did not show any benefit in preventing major cardiovascular events in healthy men in the Physicians’ Health Study II [77]. Montero et al. [78] re-analyzed the data of ten randomized controlled trials with 296 patients with T2DM and found no meaningful improvement of endothelial dysfunction caused by insulin resistance in response to a combined supplementation of vitamin C (800–3000 mg/d) and E (537–811 mg/d α-TOH). However, the authors pointed out that the body mass index (BMI) could be a potential factor impacting the effect of antioxidant supplementation in patients with T2DM [78]. Furthermore, another meta-analysis of 78 studies including 296,707 study participants suggested that using vitamin E solely or in combination with other antioxidants including vitamin C is associated with significantly increased mortality [69].

In summary, observational studies demonstrate that the combination of vitamins E and C may have a positive impact on human health by reducing oxidative stress and the risk of metabolic diseases. Despite the promising effects found in observational studies, interventional studies showed different or even controversial results (Figure 2). The reason for this could be the different types and doses of vitamin supplementation, the criteria for cohort selection, and the duration of the intervention which differed strongly between the studies. The data also suggests that age, as well as health status, affect the individual requirement of antioxidants.

#### 3.1.2. Vitamin A

The term vitamin A comprises a group of lipophilic molecules including retinol, retinal, retinoic acid (RA), as well as several carotenoids as the so-called pro-vitamin A. Vitamin A is involved in various processes in our body: it is essential for (i) vision—vitamin A deficiency can lead to night blindness, (ii) growth and development—vitamin A is involved in the genetic regulation of cell and tissue formation, and (iii) immune function [16,79]. Furthermore, as described by Huang et al., (iv) vitamin A regulates the formation of epithelial tissues, which act as the first barrier against pathogen invasion [16]. Vitamin A also plays a crucial role in the modulation of cell proliferation, differentiation, and programmed cell death of innate immune cells. Next, RA is involved in T and B cell homeostasis. Therefore, the potential role of vitamin A in the pathogenesis of COVID-19 has been also discussed [80,81].

The antioxidant properties of retinol are significantly lower than those of α-TOH [82]. As the metabolic precursor forms of vitamin A, carotenoids serve as important antioxidants in the human body. It is believed that carotenoids act as scavengers of singlet oxygen and peroxyl radicals [83]. In vitro studies revealed that there is a cooperative interaction between β-carotene and α-TOH [84]. Similar to the mechanism of the interaction of vitamin C and TOH, β-carotene may reduce the tocopheroxyl radical to TOH to restore its antioxidant properties. In the following, the generated carotenoid radical can be regenerated by vitamin C (Figure 1) [84]. However, interventional studies provided only limited evidence that β-carotene has beneficial effects on CVD [85,86]. A meta-analysis re-analyzing the data of eight randomized trials even found that β-carotene may slightly, but significantly, increase cardiovascular mortality [87]. Only few studies have investigated whether beneficial effects may exist using a co-supplementation of vitamin E and β-carotene. Upritchard and co-workers reported dose-dependent positive effects on antioxidant status in healthy subjects after consuming a vitamin E (43 or 111 mg/d α-TOH equivalents) and β-carotene (0.45 or 1.24 mg/d carotenoids)-enriched spread for eleven weeks [88]. Effects on major cardiovascular events, as well as mortality of T2DM, have been investigated as part of the ATBC trial. A total of 29,133 middle-aged male smokers were supplemented with 50 mg/d α-TOH and 20 mg/d β-carotene for a median of 6.1 years. No protective effect has been observed for fatal CVD and diabetes mortality [89,90]. 

Due to the synergistic interaction between antioxidants with vitamin function, the supplementation with a mixture of vitamin E, β-carotene, and vitamin C was tested in the prevention of oxidative stress-induced diseases including cancer, inflammation, and metabolic disorders such as CVD and T2DM [103,104,105,106,107]. Previous reviews have summarized the results of several large-scale randomized controlled trials in which the effect of supplementations with multiple antioxidant vitamins was studied. They found that the risk of major cardiovascular outcomes is not significantly changed by the supplementation of multiple antioxidant vitamins including vitamin E and β-carotene [108,109].

In summary, the majority of high-quality clinical intervention trials indicate consistently that supplementation with vitamin E and β-carotene has reliable beneficial effects on CVD (Figure 2). Some studies demonstrated that β-carotene may be associated with increased mortality on cardiovascular events and lung cancer [103]. A subsequent analysis of ATBC found that the xenobiotic metabolites of subjects were changed significantly in response to β-carotene supplementation. This suggests that long-term supplementation may affect CYP450 activity. The authors hypothesized that the underlying mechanism of induced mortality caused by β-carotene could be associated with the induced xenobiotic metabolism via CYP450 enzymes [110]; however, more studies need to be conducted to fill the evidence gap. 

#### 3.1.3. Vitamin K

The term vitamin K comprises the molecules phylloquinone (vitamin K_1_), as well as the menaquinones (vitamin K_2_ or MKs). Vitamin K functions as a cofactor for the enzyme glutamyl carboxylase that is involved in blood coagulation, bone metabolism, and prevention of vessel mineralization, as well as many other essential cellular functions [111]. Disordered mineralization in blood vessels leads to an increased risk of CVD. Vitamin K deficiency is associated with accelerated coronary artery calcification [112]. Observational studies also suggest that vitamin K has a potential role in maintaining cardiovascular health, especially in the population subgroup at increased risk of CVD and with chronic kidney disease [113]. The interventional studies investigating the impact of vitamin K on cardiovascular endpoints are too scarce to show any beneficial effect in the prevention of CVD [113,114]. However, Linneberg and colleagues reported that a low vitamin K status was associated with increased mortality in patients with COVID-19, but underlying mechanisms remain unclear [8]. Another study revealed that the vitamin K-dependent enzyme matrix gla protein (MGP), which inhibits elastic fiber calcification, is involved in the progression of respiratory failure due to COVID-19 [9]. An interventional study is needed to unravel the role of vitamin K itself as well as in combination with other nutrients in improvement of COVID-19 outcomes. 

The interference of vitamin K with vitamin E, which predominantly affects blood clotting, is known for decades and has been shown in various supplementation studies in rats [115] and humans [116,117]. A large clinical intervention study exemplifying this interaction is the “Women’s Health Study”, a randomized controlled trial to study the efficacy of vitamin E supplementation for the prevention of CVD and cancer [117]. Almost 40,000 healthy women older than 45 years received either 600 IU *RRR*-α-TOH or placebo every other day during a ten-year study period. Here, vitamin E supplementation significantly reduced major cardiovascular events by 26% and cardiovascular death by 49% [117], possibly due to reduced risk of blood clot formation mediated by vitamin E. Consistent with this hypothesis, an increase in nosebleeds was observed in the vitamin E group, and a follow-up analysis of the Women’s Health Study revealed that vitamin E supplementation decreased venous thrombus formation by 21% [116,117]. Interestingly, the correlation between high-dose intake of vitamin E and an enhanced bleeding tendency was for the first time explained in a rat model. Here, hypoprothrombinemia induced by high-dose supplementation of vitamin E could be suppressed by additional vitamin K supplements, indicating that the interference of vitamin K activity by vitamin E leads to the observed increase in bleeding tendency [115]. Thus, the vitamin E supplements in the Women’s Health Study probably acted as an anti-thrombotic agent, resulting in a reduction of clot formation by suppression of vitamin K activity [118]. Nevertheless, the underlying mechanisms remain unknown, but recent research has shed new light on this issue. Based on their joint use of enzymes of the hepatic metabolic pathway, including the initial ω-hydroxylation followed by subsequent β-oxidation, an interaction on the metabolic level is most likely [21,119]. Maret Traber [118] devised three hypothetic mechanisms for a metabolic interaction of vitamins E and K: (i) vitamin K_1_ must be converted to MK4, which is the most active extrahepatic metabolite of vitamin K by truncation of the K_1_ side-chain and replacement with a geranylgeranyl residue. Vitamin E may compete with vitamin K_1_ for the yet unconfirmed enzyme that truncates the side-chain of vitamin K_1_, thereby interfering with the initial metabolic activation of vitamin K_1_. This concept is supported by a recent observation of Hanzawa et al. in a rat model [120]. In this study, rats were supplemented with 0.75 mg/kg vitamin K_1_ for six weeks. The simultaneous application of different amounts of α-TOH (10 mg/kg to 500 mg/kg body weight) resulted in a reduction of phylloquinone concentrations in all analyzed extrahepatic tissues. The experiment has been repeated with a diet containing 0.75 mg/kg MK4 instead of K_1_. Here, the simultaneous application of α-TOH did not affect the concentration of menaquinone in extrahepatic tissues [120]. This observation provides evidence that α-TOH interferes with an intermediate step in the formation of tissue-specific MK4 from phylloquinone. (ii) The metabolic degradation of both vitamins starts with an initial ω-hydroxylation by CYP4F2; vitamin E may compete with vitamin K_1_ for the catalytic capacity of CYP4F2, thereby interfering with the formation of MK4 by preventing β-oxidation of the side-chain of vitamin K_1_ [118,119,121]. However, observations of a more recent study make this hypothesis seem less likely. Farley and co-workers showed in the cell-free system that the catalytic efficiency of CYP4F2 for the hydroxylation of phylloquinone is higher than that of α-TOH, while the co-incubation of both compounds did not influence the metabolism of phylloquinone [122]. Thus, this study indicates that the activity of CYP4F2 is not enhanced by high concentrations of α-TOH. (iii) Vitamin E increases the expression and activity of enzymes involved in its own degradation; thus, it is possible that the metabolism of vitamin K is also enhanced under elevated vitamin E concentrations [21,118]. This may force higher excretion rates of vitamin K, resulting in vitamin K deficiency with enhanced bleeding risk [111]. In line with this hypothesis, Hanazawa and co-workers showed that a 40-days diet with a content of 20% (*w*/*v*) sesame seeds (containing high amounts of sesamin as a natural inhibitor of CYP4F2 as well as CYP3A4 which catalyze the enzymatic vitamin E degradation in hepatocytes [123,124]) increased the plasma concentrations of γ-TOH, vitamin K_1_ and MK4 in male Wistar rats compared to animals fed with a control diet without sesame seeds [125]. In another investigation to study the influence of α-TOH on vitamin K metabolism and excretion, rats were fed a vitamin K-enriched diet (2 μmol/kg/d a mixture of vitamin K_1_ and K_3_) for 2.5 weeks. Then, rats were injected with a daily dose of 100 mg/kg body weight α-TOH for ten days, leading to a significant decrease in vitamin K concentration in all measured extrahepatic tissues [126]. 

There is also increasing interest in determining the contribution of vitamin K to the improvement of metabolic diseases such as cardiovascular and kidney diseases [127,128,129]. Vitamin-K-dependent proteins, such as MGP, have been observed to positively correlate with increased vascular calcification, which aggravates cardiovascular complications and mortality [127]. Considering the current evidence for an interference of vitamin E with the metabolism of vitamin K, the finding that high-dose vitamin E intake suppresses vitamin K status and may consequently have negative impact on physiological processes requires further verification and investigation.

### 3.2. Minerals

#### 3.2.1. Magnesium

Magnesium is the fourth most abundant mineral in the body, with the highest content in bones, followed by muscles, soft tissues, and blood. Seeds, nuts, and green leafy vegetables, such as spinach, are rich dietary sources of magnesium [130]. Over 75 years ago, it was first reported that a deficiency of magnesium may be associated with a pro-inflammatory state [131]. After that, more evidence has been provided on the connection between magnesium deficiency and numerous metabolic diseases. Magnesium deficiency is a risk factor for diseases connected to inflammation and oxidative stress, including obesity, atherosclerosis and T2DM [130]. The plasma concentration of magnesium negatively correlates with the risk of many metabolic diseases which are accompanied by low-grade inflammation. Animal and human studies also support that CVD patients have a significantly higher requirement of both magnesium and vitamin E [132,133]. Like vitamin E, supplementation of magnesium shows different but beneficial effects in regulating glucose, as well as lipid metabolism [44,45,134]. For this reason, it has been hypothesized that the combined supplementation could probably provide more benefits than the supplementation of magnesium or vitamin E alone. 

Four interventional studies were performed in Iran to determine the effects of a co-supplementation of magnesium and vitamin E on oxidative stress, cardiovascular risk, T2DM as well as blood lipid profile [98,99,100,101]. All of them used the same study design: the participants received 250 mg/d magnesium and 268 mg/d α-TOH. Afzali and co-workers reported that this co-supplementation of magnesium and α-TOH for twelve weeks reduced the wound size in patients suffered from diabetic ulcer [98]. In addition, taking magnesium and vitamin E led to improved glycemic control via a reduction of fasting plasma glucose and alleviative insulin resistance. The supplementation also slightly decreased triglycerides (TG) and LDL-cholesterol as well as reducing high-sensitivity C-reactive protein (hs-CRP) by more than 30%. Maktabi et al. [100] investigated the effects in patients with gestational diabetes and found that the co-supplementation ameliorated the adverse plasma lipid parameters: plasma TG and the ratio of total cholesterol to high-density lipoprotein (HDL)-cholesterol were reduced after six weeks. In the study of Jamilian et al. [100], the modulation of cardiometabolic risk, biomarkers of inflammation and oxidative stress in patients with polycystic ovary syndrome (PCOS) by the co-supplementation of magnesium and vitamin E was evaluated. Jamilian and colleagues [100] reported that in women with PCOS, the co-supplementation showed positive effects on glycemic control and reduced the cardiometabolic risk by lowering hs-CRP and plasma lipid concentrations. The study of Shokrpour et al. [99] revealed that the co-supplementation was associated with improved oxidative stress markers such as total antioxidant capacity (TAC) in these patients. Nitric oxide (NO) is an cellular signaling molecule modulating vascular tone, whose circulating concentration declines in patient with PCOS [135]. NO concentrations were increased by 10% after co-supplementation compared to the placebo, which indicated an improved endothelial dysfunction in PCOS. A meta-analysis evaluated the data of all four studies and concluded that taking magnesium and vitamin E together can reduce cardiovascular risk by improving glycemic and lipid status. However, the benefit on glucose metabolism was not significant and is probably of no clinical relevance [136]. 

In conclusion, clinical trials suggest that co-supplementation of vitamin E and magnesium may have beneficial impact on glycemic control and cardiovascular risk (Figure 2), but the existing studies have some limitations. Firstly, in all interventional studies, only the concentration of magnesium in plasma was confirmed after the period of supplementation but the concentration of vitamin E was not. Therefore, the association between the bioavailability of vitamin E and the observed effects cannot be evaluated. Further, control groups for individual supplementations of either vitamin E or magnesium are lacking in all trials. Thus, these studies do not provide clear evidence that the co-supplementation of vitamin E and magnesium has stronger effects compared to the individual supplementation of either micronutrient. Farvid and co-workers [137,138,139] analyzed the impact of minerals (magnesium and zinc), antioxidant vitamins (vitamin C and E), and their combination on T2DM patients. The study showed that the combination of these micronutrients significantly improved blood pressure, blood lipids and complications of diabetes including diabetic glomerular dysfunction compared to supplementation with only vitamins or minerals alone [137,138,139]. This finding indicates that the combination of vitamin E and minerals may be more advantageous for clinical use. 

The molecular mechanism underlying the proposed interaction of vitamin E and magnesium remains unclear. The study of Nakanishi et al. indicates that magnesium ions may be involved in kinetic stabilization of the tocopheryl radical [140]. The observation demonstrated that a direct interaction between magnesium and vitamin E may contribute to maintaining the concentration of vitamin E in the intracellular space. 

In conclusion, the number of studies focusing on the effect of a co-supplementation of vitamin E and magnesium is low. To the best of our knowledge there is no study that compares the effect of vitamin E or magnesium supplementation alone with the effect of a co-supplementation. However, the co-supplementation seems to provide some benefits to patients with impaired glucose metabolism and increased cardiovascular risk.

#### 3.2.2. Selenium

Selenium is a potent immune modulator and antioxidant micronutrient [141]. It is present most abundantly in some nuts (Brazil nut), whole grains, dairy products, animal viscera, and sea animals. The mechanisms by which selenium affects the immune system and redox systems rely on the incorporation of selenium ions into proteins known as selenoproteins [142]. The best-characterized selenoproteins include GPxs, thioredoxin reductases, iodothyronine deiodinases, methionine sulfoxide reductases (MSRs), selenophosphate synthetases, and selenoprotein K and have been reviewed extensively by Avery and Hoffmann [142].

Selenoproteins play an essential role in the production of antibodies, as shown in a study with mice lacking selenocysteine tRNA in their T cells [143]. A study conducted in eleven healthy men of an average of 31 years of age, has shown the effect of adequate and deficient dietary selenium intake on immune reaction: in this study, a selenium-rich diet increased antibody titers against diphtheria vaccine [144]. In a study with macrophages, selenium induced a phenotypic switch from the classically activated, pro-inflammatory phenotype (M1) to the alternately activated, anti-inflammatory phenotype (M2) [145]. Selenium and selenoproteins also regulate the migration and phagocytosis of macrophages [146,147]. Selenium protects neutrophils from endogenous oxidative stress [148], and increases the cytotoxic activity of natural killer cells [149]. 

ROS can oxidize methionine residues in proteins to form sulfoxides that can be reduced by MSR [150]. A well-known member of the MSR family, MSRB1, contributes to the positive regulation of macrophage actin polymerization by reducing methionine sulfoxides in actin back to methionine. Dynamic actin polymerization is a very salient phenomenon for macrophages, as it is required for phagocytosis and secretion of cytokines, two processes that are critical for mounting an immune response [151]. This is an example of a general mechanism by which redox reactions, and, by extension, antioxidants, regulate the functions of components of the cellular immune system.

Notwithstanding, not all studies support the idea of boosting the immune system by consuming more selenium, as conflicting results were obtained from clinical trials in humans [141]. Many studies carried out in humans used a combination of supplements including selenium, rather than selenium alone. The nutrients that have been used mostly in combination with selenium include zinc as well as the vitamins A, C, and E [106,152,153,154,155,156,157]. Only a few clinical trials have been carried out to study the effect of selenium in combination with vitamin E alone. However, there is a multitude of studies on this specific combination in animals, such as rats or chickens [158,159,160,161]. 

The mechanisms by which selenium and vitamin E interact to inhibit ROS-mediated oxidation have been studied in detail. In summary, there seems to be a compensatory role played by both, selenium, and vitamin E, whenever there is a deficiency of one of them. This is likely because they can both reduce oxidized lipids. In the case of vitamin E, this is a direct reaction, whereas for selenium it is indirect, as selenium is involved in the form of selenoproteins such as GPx and thioredoxin reductase [162]. Both micronutrients carry out their reducing actions exclusively, probably based on which micronutrient encounters the prooxidant first. It is well known that when vitamin E reduces lipid peroxides, it is converted into a prooxidant that has to be reduced by another antioxidant such as vitamin C, leading to the formation of ascorbate radical. The ascorbate radical is subsequently reduced by glutathione, present in selenium-dependent GPx (Figure 1) [163,164]. This indirect interaction between vitamin E and selenium is likely to be part of the rationale behind clinical trials that are carried out with a combination of more than two antioxidant micronutrients. In the study by Fisher et al. [165], genes in the liver of rats were identified, whose expression was altered in a situation of combined but not individual deficiency of vitamin E or selenium. These genes include multi-specific organic anion transporter (cMOAT) and serine protease inhibitor 3 (serpin-3), which are involved in the inflammatory response, as well as α-1-acid glycoprotein and metallothionein-1 which are involved in the acute phase of the inflammatory process. In addition, the authors observed a significant down-regulation in the expression of genes involved in inhibiting apoptosis, such as defender against cell death 1 protein (DAD1) and B-cell lymphoma-2-like protein 1 (Bcl2-L1), a protein for cell cycle regulation and antioxidant defense [165].

Animal studies revealed encouraging results, since experimental conditions are much easier to control than in clinical trials. In a streptozotocin model of diabetic rats, a co-supplementation of vitamin E (60 mg/kg/d) and sodium selenite (1 mg/kg/d) led to a reduction in blood glucose concentrations, alongside an increase in the activities of antioxidant enzymes including SOD, catalase and GPx [166].

It has been shown that a combined deficiency of vitamin E and selenium results in severe liver damage and cell death of hepatocytes in both rats and mice [167]. It is therefore believed that this combined micronutrient deficiency is a trigger for the progression of liver cirrhosis [167]. Moreover, decreasing plasma concentrations of selenium correlate with the progression of non-alcoholic fatty liver disease (NAFLD) [167], and metabolic syndrome in humans [156,168]. Next, there is an association between a diet deficient in vitamin E and NAFLD in humans, with vitamin E interventions leading to improved outcomes [169]. Hence, the effect of vitamin E and selenium on metabolic syndrome and the progression of NAFLD to non-alcoholic steatohepatitis (NASH) was studied in hamsters. This was done by combining a classical high-fat and cholesterol-rich diet with a combined deficiency of selenium and vitamin E [167]. S-adenosyl methionine (SAM) is a biomarker for the progression of fatty liver disease. In the aforementioned study, NASH was successfully triggered in hamsters using a combined deficiency of selenium and vitamin E in combination with a high-fat diet, and hepatic concentrations of SAM were higher than in livers of hamsters fed with a high-fat diet only. This study clearly indicates the key role of selenium and vitamin E in the homeostasis of SAM and in the progression of fatty liver disease [167]. A study carried out in rats by Hussein and co-workers [170] tested the effect of a combination of selenium and vitamin E on kidney function. Intriguingly, the study showed that 2 mg/kg/d sodium selenite for a duration of two to four weeks can be toxic to the renal cortex, whereas a co-supplementation of 2 mg/kg sodium selenite and 100 mg/kg/d vitamin E in form of α-TOH revealed an ameliorative function for vitamin E. In another study, the combination of 0.5 mg/kg/d sodium selenite and 100 mg/kg/d vitamin E in the form of α-TOH for three weeks protected the liver of rats from the toxic effects of the xenobiotic bisphenol A [171]. A combination of 0.25 mg/kg/d sodium selenite and 100 mg/kg/d vitamin E reportedly protected Wistar rats from bifenthrin-induced atherogenesis [172]. In another study in rats, a combination of 20 mg/d vitamin E and 0.3 mg/d selenium protected the heart from prednisolone-induced lipid peroxidation [173]. Several studies have also been carried out in chickens to test the effect of a dietary supplementation of a combination of selenium and vitamin E on immune responses. In a study with chickens, Dalia et al. [159] compared the effect of a co-supplementation of 100 mg/kg/d vitamin E (α-TOH acetate) and 0.3 mg/kg/d selenium of bacterial origin, with a supplementation of either one of the two micronutrients. A synergistic effect of the co-supplementation was observed on plasma levels of immunoglobulins and cytokine expression in the spleen of the chickens. The finding is significant that the synergistic effect only occurred when selenium originating from bacteria (as opposed to inorganic sodium selenite) was used [159]. The synergistic effect on immunoglobulins observed in this study has been confirmed in studies by Singh et al. as well as Swain et al. [160,161]. 

In contrast, clinical trials in humans revealed contradictory results. In a clinical trial with 170 patients suffering from T2DM, hypertension and/or microalbuminuria [95], the patients were randomized into a control group, a group receiving 268 mg/d vitamin E, a group receiving 200 μg/d selenium and a group receiving a combination of the two micronutrients with the aforementioned doses. After a twelve-month intervention, the treatment with vitamin E and selenium led to improved circulating concentrations of selenium-dependent GPx3. A restoration of glomerular filtration and improved renal blood flow was also observed in the group with the combined supplementation compared to the control group [95]. This study indicates that long-term co-supplementation of selenium and vitamin E could probably elicit protective effects on kidney function in patients with metabolic disorders such as T2DM, hypertension and microalbuminuria. 

In a recent clinical trial, it was shown that women suffering from premature ovarian insufficiency benefited from a 90-day supplementation with 200 μg/d selenium and 268 mg/d vitamin E [96]. In this study, 70 women of Iranian origin ranging from 20 to 40 years of age were randomized into only a placebo and a treatment group; therefore, no data on the individual effect of each micronutrient are available. Nevertheless, the evaluated fertility parameters including anti-Mullerian hormone, antral follicle count, and mean ovarian volume was improved significantly in the supplemented group, indicating that the two micronutrients, act together, likely by combating ROS [96]. 

The effect of long-term supplementation (36 months) of a combination of 268 IU/d vitamin E and 200 μg/d L-selenomethionine, as well as the individual micronutrients on oxidative stress was investigated in 312 male smokers who were a subset of participants from the SELECT study [97]. The participants were randomized into four groups as follows: control, vitamin E only, selenium only, and a combination of vitamin E and selenium. The study revealed that when compared to the placebo group, vitamin E alone reduced the level of the oxidative stress marker 8-*iso*-PGF2α in urine. On the other hand, the combined treatment of selenium and vitamin E as well as selenium alone did not cause significant changes [97]. 

Differences in dosage and/or source of the micronutrient supplements, as well as the wide range of study participants differing, for example, in age, ethnicity, and gender, makes it difficult to compare the results of the clinical trials conducted so far. Nevertheless, this may explain the diversity of results gotten from studies investigating the effects of a combination of selenium and vitamin E.

#### 3.2.3. Zinc

Zinc is an essential micronutrient whose deficiency correlates strongly with an impaired immune response [174]. Main sources of zinc are seafood, cereals, turkey, and lentils [175]. Even though zinc itself is a redox-inert metal, it plays a role as an antioxidant by aiding the proper functioning of the copper/zinc-containing antioxidant enzyme SOD. This enzyme catalyzes the reaction in which superoxide radicals generated from metabolic reactions in the body get converted to hydrogen peroxide, which is then metabolized to oxygen and water by the enzyme catalase [174]. Zinc also plays an antioxidant role in the maintenance of sulfhydryl groups of proteins, the stabilization of membranes, and upregulation of metallothionein expression (Figure 1) [174]. 

Metallothioneins comprise a group of proteins containing numerous cysteine residues, with which they bind metal ions including zinc, selenium, and copper. With their ability to bind metal ions, Metallothioneins contribute to the prevention of oxidative stress caused by heavy metal toxicity. These proteins also prevent oxidative stress by scavenging free radicals directly with their cysteine groups. The zinc/sulfur cluster bound in metallothionein acts as a redox unit to regulate the cellular redox state (reviewed by Maret in [176]). Zinc also contributes to the expression of metallothioneins, by binding to respective response elements in the DNA (reviewed by Krezel and Maret in [177]). In addition, zinc binds to several proteins that are crucial for inflammation such as tumor suppressor phosphatase and tensin homolog (PTEN) and STAT3, thereby disrupting their activity [178]. The mechanisms by which zinc alters the response during infection, inflammation and immunity have been comprehensively reviewed [178,179], and are also schematically illustrated in Figure 1.

Vitamin E also induces the expression of metallothioneins in cardiomyocytes of rats exposed to heat stress, with a consequent restoration of mitochondrial function and reduction in oxidative stress, as evidenced by increase in SOD and GSH and a reduction in MDA and lactate dehydrogenase [180]. Vitamin E also regulates the activity and expression of Cu/Zn-SOD in the blood of rats [181].

Since there is evidence that both zinc and vitamin E act as strong antioxidants and share some mechanisms of action, a few studies have been aimed at understanding the effect of a co-supplementation of zinc and vitamin E or of a combined deficiency of these nutrients. However, most of these studies were performed in animals rather than humans. In 2007, Bruno and co-workers investigated the effect of zinc deficiency on oxidative stress and antioxidant status [182]. Zinc deficiency caused a reduction of hepatic α-TOH and γ-TOH concentrations, and an elevation in plasma F_2_ isoprostanes, biomarkers of oxidative stress, compared to the rats fed a zinc-adequate diet [182]. Interestingly, an increase in hepatic CYP4F2, the enzyme involved in tocopherol catabolism, was observed in the zinc-deficient rats [182]. The authors could not provide data on the mechanism by which zinc affects vitamin E metabolism and/or function but hypothesized that zinc may affect the bioavailability of vitamin E: the deficiency of zinc may cause oxidative stress and depletion of the available vitamin E because it is oxidized in the process of combating peroxyl radicals. Zinc also plays a role in maintaining sulfhydryl groups in proteins including GSH [182]; thus, zinc deficiency leads to a depletion of GSH which is responsible for reducing vitamin C that is in turn responsible for restoring vitamin E to its reduced form (Figure 1) [164,182]. Only few clinical trials have been conducted to study the effects of a co-supplementation of vitamin E and zinc, but several have been conducted that investigate a co-supplementation of additional antioxidant micronutrients such as vitamin C [183], vitamin A, and selenium [184], as well as copper [185]. This is likely because different antioxidants work together in vivo to combat oxidative stress. Therefore, an interdependence of these antioxidants, as well as overlaps in their mechanisms of action exist (Figure 1). This indicates the importance of a balanced diet, containing the appropriate amounts of these micronutrients.

A clinical trial studied the effect of a co-supplementation of zinc and vitamin E on the metabolic status of 54 Iranian woman aged between 18 to 40 years and suffering from gestational diabetes mellitus [102]. Study participants were randomized into two groups: placebo or a supplementation with 233 mg/d zinc gluconate and 268 mg/d α-TOH for a duration of six weeks. The co-supplementation of zinc and vitamin E caused a significant drop in serum insulin levels as well as serum total cholesterol and LDL-cholesterol [102]. This group also had higher insulin sensitivity indicating an improved health state of the gestational diabetic patients, and showed an upregulation of the expression of peroxisome proliferator-activated receptor γ (PPARγ) and the LDL receptor compared to the placebo group [102]. A drawback of this study is that there are no groups in which zinc and vitamin E were supplemented alone, so that it is difficult to distinguish whether the observed effects are caused by the combination of these two micronutrients or one nutrient alone. Interestingly, in a study by Karamali and colleagues four years earlier, a supplementation of zinc was conducted in comparison to placebo [186]. In this study, featuring 58 primigravid Iranian women aged between 18 and 40 years suffering from gestational diabetes mellitus, the supplementation of 30 mg/d zinc for six weeks also led to lower fasting blood glucose and blood insulin concentrations, higher insulin sensitivity, and lower plasma TG. It is, therefore, possible that in the co-supplementation study of Ostadmohammadi et al. [102] zinc played the major role for the observed results. However, due to the lack of respective control groups with the individual nutrients in this study, it is not possible to draw a conclusion. Therefore, further clinical studies are needed to better understand the effects of a co-supplementation of zinc and vitamin E.

## 4. Biological Activities of Long-Chain Metabolites of Vitamin E

Although vitamin E is mainly known for its antioxidant function, there is growing evidence that vitamin E interacts with several regulatory proteins, thereby modulating signaling pathways as well as gene expression (reviewed by Rimbach et al. in [187,188,189]). Clinical trials studying the pharmacological effects of high-dose vitamin E supplementation on diseases with inflammation as an underlying mechanism, such as atherosclerosis, diabetes, and cancer, have not been convincing, since some trials produced results suggesting that vitamin E has no pharmacological benefit on health or may even increase the risk for chronic diseases and mortality [33,41,188,190,191,192,193]. Such unexpected results indicate that there may be mechanisms underlying the biological activities as well as the metabolism of vitamin E, which are not fully understood yet. This has led to the concept that vitamin E, just like its fellow lipid-soluble vitamins A and D, may require an activation to a metabolite that elicits signaling activities that are more potent and sometimes different from that of the precursor [27]. The metabolism of vitamin E has been mostly studied in animals but its long-chain metabolites (LCMs) have also been found in human serum [31,32], providing evidence for their systemic bioavailability. These findings also indicated that these metabolites may exhibit physiological functions, not only in the liver, but also in extra-hepatic tissues. Since then, several studies have been carried out by different research groups to unravel the signaling pathways that are regulated by the LCMs. Attention has also been paid to the dose-response and differential signaling of the metabolites compared to their precursors, with several reports showing that the bioactivity of the α-TOH-derived LCMs occurred at lower concentrations and with mechanisms distinct from those of their metabolic precursor, α-TOH [31,32].

The endogenously formed LCMs elicit strong anti-inflammatory activities [31,32,33,194,195,196]. In vitro studies carried out in murine macrophages revealed that the LCM α-13’-COOH suppresses the proinflammatory enzymes cyclooxygenase 2 (Cox2) and iNos and consequently reduces the formation of NO and the secretion of proinflammatory cytokines more efficiently than α-TOH [196]. Furthermore, the LCMs affect the expression of several genes, subsequently altering cellular processes including scavenger receptor CD36-mediated phagocytosis in human macrophages [31,32]. Pein et al. [33] showed that α-13’-COOH accumulates in human leukocytes, inhibits the inflammation-causing 5-lipoxygenase in human polymorphonuclear leukocytes, and also relieves acute inflammation as well as bronchial hyper-reactivity in a murine model of peritonitis and asthma. Our group has also recently discovered that α-13’-COOH suppresses the expression of the proinflammatory chemokine CC-chemokin-ligand-2 (CCL2) in murine macrophages and therefore exhibits anti-inflammatory effects via MAPK and NF-κB pathways [194].

Due to the relevance of lipid metabolism for the development of metabolic diseases, the effects of the LCMs on lipid homeostasis have been investigated. Our group has shown that α-13’-COOH protects THP-1 macrophages from stearic acid-induced lipotoxicity by increasing the expression of the lipid droplet-associated protein (PLIN2, adipophilin), in contrast to its precursor α-TOH which exerts opposite effects [34]. Recently, Kluge et al. [35] reported that α-13’-COOH reduces foam cell formation of human THP-1 macrophages via an increase in the expression and release of angiopoietin-related protein 4 (ANGPTL4), which reduces TG hydrolysis from TG-rich lipoproteins such as very-low-density lipoproteins (VLDL) by inhibiting lipoprotein lipase (LPL) activity.

The structure–function relationship of the LCMs has also been investigated by comparing partial and full forms of the LCM structures to their parent compounds. The results showed that, at least in vitro, both the chromanol ring and the modified side-chain together confer the distinct biological functionality to LCMs [34]. This study provided more evidence that the LCMs elicit specific and distinct signals from that of their precursor compounds. 

Garcinoic acid (also known as δ-13’-tocotrienol-carboxychromanol) is, in principle, the LCM of δ-tocotrienol (Figure 3) and is abundant in seeds of *Garcinia kola*. It has been particularly well studied since it elicits generally strong effects and is not as promiscuous as other LCMs [36]. Garcinoic acid has been found to exhibit anti-inflammatory effects in activated murine macrophages. Natural isoforms of garcinoic acid as well as semi-synthetic isoforms have been shown through molecular docking and the use of human A549 fibroblasts to be inhibitors of microsomal prostaglandin E_2_ synthase 1 (mPGES-1), which partially explains the anti-inflammatory effects of this compound [197]. Willems et al. [36] tested garcinoic acid, as well as α-carboxy- and α-hydroxy-LCMs, for their ability to activate PPARγ signaling. α-13’-COOH and garcinoic acid, in that order, were found to be comparatively potent in activating PPARγ than α-hydroxy-LCMs. An allosteric mode of co-regulation of PPARγ and an orthosteric mode were characterized as part of the mechanisms by which garcinoic acid activates PPARγ signaling [36]. These results suggest that PPARγ activation by the LCMs may be a part of the mechanisms involved in the pharmacological functions of vitamin E. The group of Qing Jiang has also shown that the LCMs and garcinoic acid inhibit the growth and induce the death of cancer cells, reviewed recently by Jiang [193]. With the aid of x-ray crystallography, cell culture, and animal studies, Bartolini et al. have shown that garcinoic acid is capable of activating the pregnane X receptor (PXR) in human intestinal and liver cells as well as in mice. Compared to its metabolic precursor δ-T3, other forms of vitamin E, such as γ-TOH and α-TOH, as well as other LCMs, garcinoic acid is stronger and more selective [189]. Recently, Olajide and co-workers reported that garcinoic acid has potential in suppressing the cytokine storm caused by the SARS-CoV-2 spike protein S1 in human peripheral blood mononuclear cells (PBMCs) via inhibition of NF-κB activation [198]. This result suggests that garcinoic acid may help to limit severe symptoms and complications associated with SARS-CoV-2 infection. 

As shown in Figure 3, α-TOH, α-13’-COOH, and garcinoic acid have similar molecular structures. Structure–activity relationship studies indicate that three main structural features in the vitamin E analogues play pivotal roles for different biological functions and the intensity of the functions is also modulated by the modifications of these domains [199,200,201]: (i) The hydroxyl group on the C6-position of the chromanol ring is indispensable for the antioxidant activity [199]. (ii) The signaling domain is involved in the activation of signaling pathways regulating inflammation and cytotoxicity [201]. The hydrophobic side-chain contributes to the distribution in membranes and lipoproteins [199]. Furthermore, several structural factors of the side-chain such as the number of carbon units, existence of double bonds, methylation pattern of the chromanol system and the oxidized modification at the end of the side-chain, have been considered to affect the effect of the molecules (reviewed in [201]).

From the evidence outlined above, it is clear that the LCMs elicit crucial signaling and gene modulatory effects in cells. Recent findings revealed that the LCMs have several similarities to vitamin E as regulatory molecules. However, for some effects, such as the regulation of cellular lipid homeostasis [31,34,35] or the modulation of the inflammatory response [196], the LCMs showed superior or even different effects compared to its precursors. For that reason, it has been proposed that the metabolic activation is a premise for the functional properties of vitamin E in humans [27,188]. However, it is totally unknown at present, how the sufficiency, or conversely, the deficiency of other micronutrients may affect the biological functions of the LCMs. In the light of our current knowledge, we therefore propose the following hypotheses: 

(i) We hypothesize that metabolic activation is essential for (many of) the physiological effects known for vitamin E [27]. As described in Section 3.1, the metabolism of both vitamin E and K is initiated with the CYP450-dependent ω-hydroxylation. This means that simultaneous intake of high doses of vitamin E and K could lead to a metabolic competition, which possibly limits the potential effect of a co-supplementation of these vitamins. Thus, using purified LCMs directly as a supplement could reduce this competition to a minimum, since the LCMs act downstream of CYP450. It is also worth mentioning that the LCMs, such as α-13’-COOH and garcinoic acid, bind to PXR and induce the expression of genes such as CYP3A4 and P-glycoprotein [189,202] to the same extent as known for vitamin E [203]. This interaction may additionally affect the bioavailability of vitamin K during supplementation with either vitamin E or LCMs.

(ii) Various studies indicate that the LCMs, especially garcinoic acid, could be promising medicinal agents for treating inflammatory diseases [40]. Some of these studies also indicate that the anti-inflammatory properties of the LCMs are stronger than that of their precursors under the same conditions [33,196]. These findings suggest that the metabolites could also work better in combination with other nutrients that have anti-inflammatory activity. Since selenium induces a switch from M1 to M2 phenotype in macrophages [146], its deficiency could mean that macrophages do not easily switch from a pro- to an anti-inflammatory state. This is an important phenomenon, as it could mean that the ability of the LCMs to inhibit the inflammatory response may depend on the availability of selenium. Ultimately, this will likely have a significant impact on the regulation of the immune response by the LCMs. It may, therefore, be for such reasons that vitamin E itself elicits contradictory effects in different clinical trials.

(iii) Since the LCMs regulate the expression of several genes, it is likely that some of these genes, or the signaling pathways in which the protein products of these genes function, are regulated by other micronutrients. For instance, some of the LCMs, γ-13’-COOH for example, have been found to regulate STAT3, which is crucial for cell proliferation and immunity amongst other functions, as well as NF-κB, which is crucial for the inflammatory response [193]. These signaling mediators are also regulated by vitamin C [204,205], retinoic acid [206], zinc [207,208], and selenium [209]. This implies that the availability of other micronutrients will likely affect the extent to which the LCMs can carry out their effects on gene regulation and cell signaling. 

(iv) So far, not much is known about the antioxidant effects of the LCMs. Nevertheless, it is safe to say that the LCMs, which also bear the hydroxyl group on the chromanol ring that is crucial for redox reactions (Figure 3), may participate directly in free radical scavenging reactions, just like α-TOH. As it is for vitamin E, this reduction may depend on the actions of vitamins A and C, as well as selenium and zinc. Therefore, the availability of these micronutrients could affect the regeneration of the proposed reduced form of the LCMs, that may be useful in maintaining the redox balance in cells. One question here is how the distribution of the different LCMs in cell compartments affects their proposed antioxidant activities. As described in Section 2, the hepatic metabolism of vitamin E occurs in three different cell compartments: endoplasmic reticulum, peroxisomes, and mitochondria. However, it is almost fully unknown how the transport of the LCMs in extra-hepatic tissues works and how the LCMs are distributed in cells of extra-hepatic tissues. All these factors may play a role in affecting the proposed radical-scavenging properties of the LCMs. All these questions remain to be answered in future studies. 

In summary, studies about the effects of micronutrients on the functions of the LCMs are warranted. Nevertheless, based on the hypotheses outlined above, experiments should be conducted to improve our understanding of the interactions between the LCMs and other micronutrients and of the effects of such interactions on human health.

## 5. Conclusions

Overall, there is growing evidence for the relevance of physiological interactions between vitamin E and other micronutrients. Vitamin C, β-carotene, selenium, and zinc likely interact with vitamin E directly and indirectly through their antioxidant capacities and functions, indicating that these interactions may act synergistically in the regulation of redox homeostasis in humans. Extensive research has been conducted to investigate the interference of vitamin E and these antioxidant micronutrients.

The interaction of vitamins E and C is considered to be important with respect to neutralizing oxidative stress. Findings from observational studies indicate that the requirements of both vitamins E and C may increase with age. Despite extensive interventional trials our knowledge on the complex synergy of these two vitamins in the prevention of metabolic diseases is still poorly understood. So far, the interfering effects of co-supplementations of vitamin E and selenium, β-carotene, and zinc, respectively, have not been confirmed by interventional trials in humans and it remains necessary to assessing whether these co-supplementations may improve the prevention of CVD, T2DM, hypertension, and other metabolic diseases. Cohort studies and randomized clinical trials revealed controversial results, indicating possible harmful effects of high-dose co-supplementations. Considering the diverse characteristics of the different trials, including differences in the selection and number of participants, dosage and duration of supplementation, as well as form of the supplements, further studies with more power, better standardization, and more complex design have to be conducted to unravel the role of the interrelation between vitamin E and other micronutrients for human health.

Magnesium and vitamin K have been reported to interfere with vitamin E metabolism. Therefore, it will be interesting to see how this may affect bioavailability and physiological functions of vitamin E. Unfortunately, only a few human studies so far deal with this matter. Studies in humans with improved study design are required to better understanding the complexity of the interaction of vitamin E and other micronutrients and its contribution to metabolic diseases.

As outlined in Section 4, the LCMs exhibit several biological effects and seem to act in a more efficient manner compared to their metabolic precursors with respect to lower effective doses and higher specificity. Next, the LCMs occur in human serum and accumulate for example in primary immune cells. Taken together, these findings support the idea that the LCMs may have regulator function, thus having opening a new perspective for research in the field of vitamin E. Our group hypothesizes that vitamin E may act—at least in part—similar to other lipid-soluble vitamins, such as vitamins A and D, which require metabolic activation before they can carry out all of their physiological functions [27]. Although only a limited number of studies is available, clear evidence has been provided for the relevant interactions of vitamin E and its metabolism with other nutrients including vitamins and minerals. It is, therefore, to be expected that these interactions affect also the endogenous formation of vitamin E metabolites and the physiological relevance in humans. Corresponding studies in cell and animal models as well as intervention trials in humans are necessary for this propose.

## Figures and Tables

**Figure 1 antioxidants-11-01785-f001:**
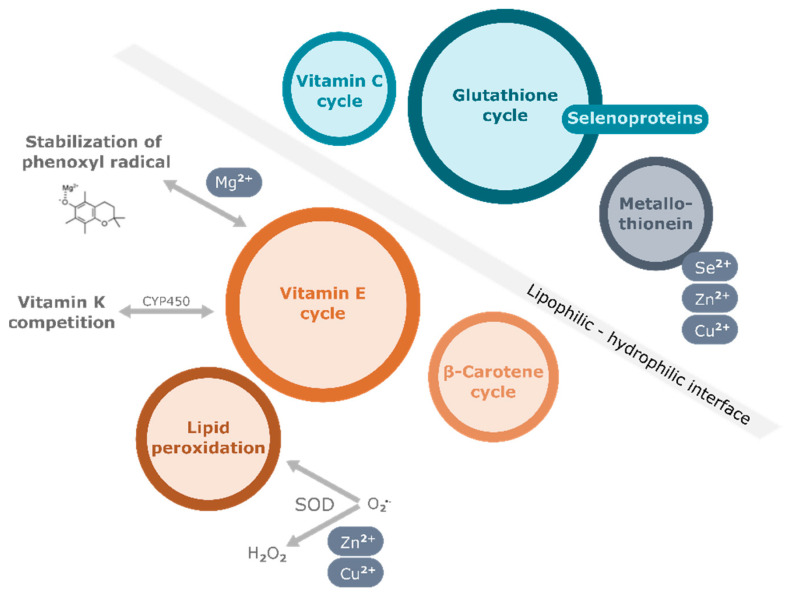
Possible mechanisms underlying the interaction between vitamin E and other micronutrients. Free radicals such as O_2_^•−^ (hyperoxide anion) attack the double bonds of the unsaturated fatty acids of membrane phospholipids resulting in the formation of a lipid peroxyl radical (RO_2_^•^). Vitamin E (TOHs and T3s) provides its phenolic hydrogen atom to the peroxyl radical and converts it to a hydroperoxide. The resulting vitamin E radical can be reduced by: Vitamin C which forms then an ascorbyl radical which is regenerated with the help of glutathione (GSH). In this process, two monomers of GSH are oxidized to form a dimer (GSSG). Selenoproteins and the metal ion transporter metallothionein are involve in reducing GSSG to GSH. Pro-vitamin A (β-carotene) regenerates TOH from tocopheryl radicals. The resulting carotenoid radical cation is further regenerated by the vitamin C cycle. Besides that, magnesium ions contribute to the stabilization of the vitamin E radical and vitamin K competes hypothetically with vitamin E for the enzyme CYP4F2 which ω-hydroxylates the side-chains of both vitamin K_1_ (phylloquinone) and vitamin E. Abbreviations: CYP, cytochrome P450; SOD, superoxide dismutase.

**Figure 2 antioxidants-11-01785-f002:**
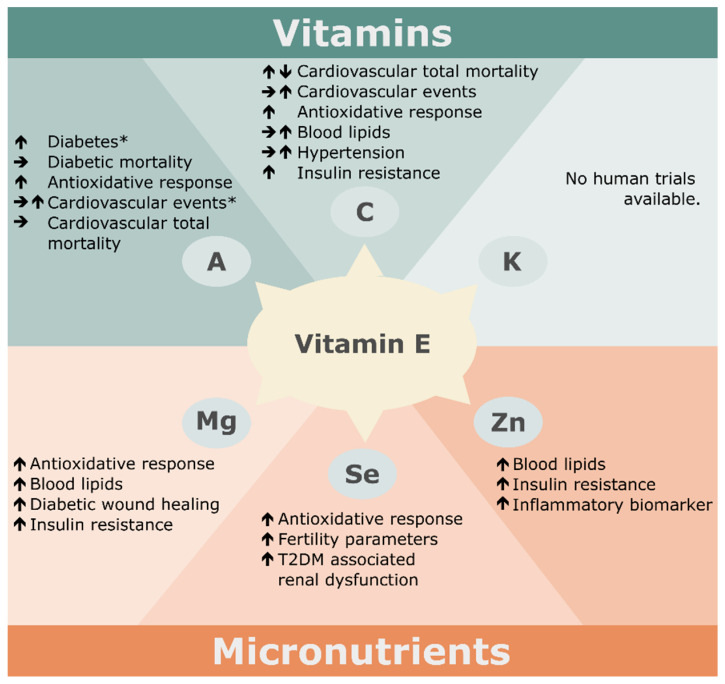
Impact of the dietary intake or the supplementation of vitamin E and other micronutrients including vitamins A (including β-carotene), C, and K, as well as zinc, magnesium, and selenium, on metabolic diseases. ↑, improved parameter after co-supplementation; ↓, deteriorated parameter after co-supplementation; →/↑, controversial results found in different trials; *, correlation shown by observational studies. Abbreviation: T2DM, diabetes mellitus type 2.

**Figure 3 antioxidants-11-01785-f003:**
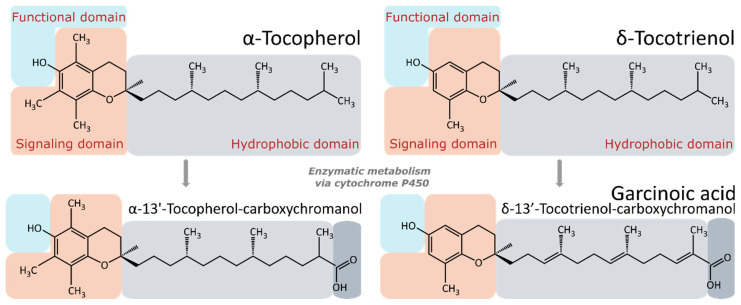
Structural similarity of α-tocopherol, δ-tocotrienol, α-13’-carboxychromanol and garcinoic acid. Three structural features can be postulated in the vitamin E analogues: the functional domain, the signaling domain and either the hydrophobic domain or the oxidized, i.e., activated side-chain.

**Table 1 antioxidants-11-01785-t001:** Co-supplementation of vitamin E and other micronutrients and their effect on metabolic diseases.

Author	Type of Study	No. of Participants	Subjects	Endpoints	Vitamin E Dosage	Co-Supplement Dosage	Duration
**Vitamin C**							
Jungert et al. [64]	Observational	399	Participants aged ≥ 60 years	Stable positive interrelation was shown between plasma concentrations of VC and VE with aging	-	-	12 years
Hemilä [66]	Observational	27,111	Smoking males aged 50–69 years	In older participants (66–69) who had VC intakes > 90 mg/d and dietary vitamin E intakes > 12 mg/d, vitamin E supplementation was significantly beneficial on mortality rate	-	-	10 years
Losonczy et al. [71]	Interventional	11,178	Participants aged 67–105 years	Reduced risk of all-cause of mortality and risk of CVD mortality in vitamin E group; lower risk of total mortality and CVD mortality by VE and VC	Use of supplement including different dosages of VE and/or VC		10 years
Hamilton et al. [63]	Interventional	30	Healthy participants	Plasma α-TOH increased with vitamin C supplementation (~10%); plasma ascorbic acid also increased with vitamin E supplementation (>12%); plasma antioxidant power and glutathione peroxidase activity increased, total cholesterol and triglyceride concentrations decreased in both supplementation groups	73.5 mg/d *RRR*-α-TOH acetate	500 mg/d VC	6 weeks
Shargorodsky et al. [91]	Interventional	70	Patients with cardiovascular risk factors	Significant decline of HbA_1C_ and increase in HDL-cholesterol were also observed by co-supplementation.	134 mg (200 IU)/d α-TOH	500 mg/d VC	6 months
Sesso et al. [77]	Interventional	14,641	Males aged > 50 with low risk of cardiovascular disease	Vitamin E/C supplementation has no effect on the incidence of major cardiovascular events; co-supplementation had no beneficial effects neither	268 mg (400 IU)/two days α-TOH	500 mg/d VC	8 years
Amini et al. [72]	Interventional	60	Females in reproductive age with endometriosis	Combined deficient intake of both vitamins was observed in 33.7% of participants before intervention; co-supplementation decreased oxidative stress marker (MDA and ROS level), no influence of TAC, improvement of endometriosis syndrome (pain, blooding) shown	537 mg (800 IU)/d α-TOH	1000 mg/d VC	8 weeks
Plantinga et al. [75]	Interventional	30	Males with untreated essential hypertension	Co-supplementation had beneficial effects on endothelium-dependent vasodilation and arterial stiffness in untreated, essential hypertensive patients; effect was associated with changes in plasma markers of oxidative stress.	268 mg (400 IU)/d α-TOH	1000 mg/d ascorbic acid	8 weeks
Mihalj et al. [92]	Interventional	57	Patients with treated hypertension	Co-supplementation showed no BP-lowering effect; not effective in reducing oxidative biomarker (8-*iso*-PGF_2α_)	537 mg (800 IU)/d α-TOH	500 mg/d ascorbic acid	8 weeks
Rafighi et al. [74]	Interventional	170	Patients with T2DM aged 30–60 years	Co-supplementation decreased blood glucose concentration, oxidative stress and insulin resistance by increasing antioxidant capacity (SOD and GST enzyme activity)	201 mg (300 IU)/d α-TOH	266.7 mg/d ascorbic acid	3 months
Mellyana et al. [76]	Interventional	42	Patients with pediatric idiopathic nephrotic syndrome aged 1–15 years	Co-supplements improved blood pressure parameters (decreased number of patients in pre-hypertensive to hypertensive stage)	10–15 mg/kg/d α-TOH	10–15 mg/kg/d ascorbic acid	12 weeks
Karajibani et al. [93]	Interventional	40	Patients with CVD	Reduced lipid peroxidation and strengthened the antioxidant defense system by (elevated SOD, GPx-activity, induced TAC and decreased MDA)	268 mg (400 IU)/d α-TOH	500 mg/d ascorbic acid	8 weeks
Bjelakovic et al. [69]	Interventional (meta-analysis)	296,707	Participants with low bias from 78 studies	Antioxidant supplements with vitamin E and C has no evidence to support primary or secondary prevention but may increase significantly mortality compared to placebo groups.	Co-supplementation including different dosages of vitamin E and C		-
**Vitamin A**							
Altoum et al. [94]	Observational	400	Healthy participants/patients with CVD/non-CVD patients with T2DM	Lower antioxidant vitamin A, E and C concentrations are associated with increased diabetes risk. For those who already had diabetes, lower concentrations of vitamin E and A were associated significantly with a higher risk of CVD.	-	-	-
Upritchard et al. [88]	Interventional	105	Healthy adults	Co-supplementation improved dose-dependently the lipid peroxidation by increasing antioxidant capacity	43/111 mg/d α-TE	0.45/1.24 mg/d carotenoids	11 weeks
Törnwall et al. [90]	Interventional	27,271	Smoking males aged 50–60 years	Co-supplementation showed no significant influence on risk of major coronary events, and no reduction of myocardial infarction and CVD mortality	50 mg/d *RRR*-α-TOH acetate	20 mg/d β-carotene	5–8 years (median: 6.1 years)
Kataja-Tuomola et al. [89]	Interventional	1700	Smoking males aged 50–60 years	Co-supplementation showed no effect on diabetes mortality during the intervention	50 mg/d *RRR*-α-tocopherol acetate	20 mg/d β-carotene	5–8 years (median: 6.1 years)
**Se**							
Zitouni et al. [95]	Interventional	170	Patients with T2DM caused renal diseases	Co-supplementation improved renal function (glomerular filtration, renal blood flow); GPx3 was shown to be involved in the restoration of renal function.	268 mg (400 IU)/d α-TOH	200 μg/d selenium	12 months
Safiyeh et al. [96]	Interventional	70	Females with OPOI	Co-supplementation improved fertility parameters (number of antral follicles and ovarian volume)	268 mg (400 IU)/d α-TOH	200 μg/d selenium	12 weeks
Guertin et al. [97]	Interventional	312	Smoking males	Co-supplementation lowered plasma biomarkers of oxidative stress (8-*iso*-PGF_2α_) in vitamin E group; no effect in vitamin E plus Se or Se only group	268 mg (400 IU)/d α-TOH	200 μg/d selenium	3 years
**Mg**							
Afzali et al. [98]	Interventional	57	Patients with grade 3 DFU	Co-supplementation improved diabetic wound healing by reduced ulcer size; decreased insulin resistance and improved glycemic control; ameliorated blood lipid profile (TG, LDL-cholesterol, HDL-cholesterol); improved inflammatory biomarker (hs-CRP); improved oxidative stress biomarker (TAC, MDA)	268 mg (400 IU)/d α-TOH	250 mg/d magnesium oxide	12 weeks
Shokrpour et al. [99]	Interventional	60	Females with PCOS aged 18–40	Co-supplementation reduced plasma inflammatory biomarker (hs-CRP); improved oxidative stress biomarker (plasma NO level; TAC) but no effect on plasma GSH and MDA	268 mg (400 IU)/d α-TOH	250 mg/d magnesium oxide	12 weeks
Jamilian et al. [100]	Interventional	60	Females with PCOS aged 18–40	Co-supplementation improved insulin metabolism (insulin concentrations, homeostatic model of assessment for insulin resistance, quantitative insulin sensitivity check index); improved blood lipide profile (TG, VLDL)	268 mg (400 IU)/d α-TOH	250 mg/d magnesium oxide	12 weeks
Maktabi et al. [101]	Interventional	60	Females with gestational diabetes aged 18–40	Co-supplementation improved glycemic control (lower blood glucose and insulin concentrations, homeostatic model of assessment for insulin resistance and higher quantitative insulin sensitivity check index); improved blood lipide profile (TG, VLDL, LDL- and total or HDL-cholesterol)	268 mg (400 IU)/d α-TOH	250 mg/d magnesium oxide	6 weeks
**Zn**							
Ostadmohammadi et al. [102]	Interventional	54	Females with gestational diabetes	Co-supplementation lowered insulin resistance (increased insulin plasma concentrations, homeostasis model of assessment; insulin sensitivity check index); improved blood profile parameter (reduced total and LDL-cholesterol); increased gene expression of lipid metabolism regulatory genes (PPARγ, LDLR)	268 mg (400 IU)/d α-TOH	233 mg/d zinc gluconate	6 weeks

Abbreviations: α-TE, α-tocopheryl acetate; α-TOH, α-tocopherol; BP, blood pressure; CVD, cardiovascular disease; DFU, diabetic foot ulcer; GPx, glutathione peroxidase; GST, glutathione S-transferase; HbA_1c_, hemoglobin A_1c_; HDL, high-density lipoprotein; hs-CRP, high sensitivity C-active protein; LDL, low-density lipoprotein; LDLR, low-density lipoprotein receptor; MDA, malondialdehyde; NO, nitric oxide; OPOI, occult premature ovarian insufficiency; PCOS, polycystic ovary syndrome; PGF_2α_, Prostaglandin F_2α_; PPARγ, peroxisome proliferator-activated receptor γ; ROS, reactive oxygen species; T2DM, type 2 diabetes milieus; TAC, total antioxidant capacity; TG, triglyceride; SOD, superoxide dismutase; VLDL, very-low-density lipoprotein; VC, vitamin C; VE, vitamin E.

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
