# Peer review of "Vitamin E and Metabolic Health: Relevance of Interactions with Other Micronutrients"

_antioxidants, 2022, doi:10.3390/antiox11091785_

Round 1
Reviewer 1 Report
· I read the manuscript with great enthusiasm. The manuscript interestingly depicts the interrelationship between vitamin E and other nutrients.
· The manuscript is well drafted and with clarity of presentation of complex functions and cellular interactions of Vitamin E with other essential micronutrients.
· Figures are representative of critical thinking and available evidence.
· As the manuscript discusses the nutrients and their metabolites, it would be good to have a few lines incorporated regarding the biomarkers of vitamin E and its usefulness in human trials in metabolic diseases. This would add additional information to the readers.
· Some information on the bioavailability of Vitamin E supplements in human trials would add exciting insights.
Author Response
Dear Editor,
Dear Reviewers,
We gratefully acknowledge your careful and kind evaluation of our manuscript “Vitamin E and metabolic health: relevance of interactions with other micronutrients” and your constructive criticism which helped us to improve the overall quality of our review. We respond in detail to all of the valuable suggestions made by the reviewers. Please find in the following a point-by-point reply to your comments. We accordingly made changes to the manuscript in response to your comments which are highlighted in red in the revised manuscript.
Yours sincerely,
Stefan Lorkowski
Reviewer 1
- I read the manuscript with great enthusiasm. The manuscript interestingly depicts the interrelationship between vitamin E and other nutrients. The manuscript is well drafted and with clarity of presentation of complex functions and cellular interactions of Vitamin E with other essential micronutrients.
REPLY: We are very pleased about the reviewer's kind assessment to our review.
- Figures are representative of critical thinking and available evidence.
REPLY: We thank the reviewer for this positive feedback on our figures.
- As the manuscript discusses the nutrients and their metabolites, it would be good to have a few lines incorporated regarding the biomarkers of vitamin E and its usefulness in human trials in metabolic diseases. This would add additional information to the readers.
- Some information on the bioavailability of Vitamin E supplements in human trials would add exciting insights.
REPLY to point 4 and 5: We thank the reviewer for this valuable suggestion to provide information on the bioavailability of vitamin E supplementation as well as the biomarkers of vitamin E status after supplementation in human trials. To address the two comments, we added the new section 2.2 bioavailability of vitamin E, which reads as follows (lines 94 to 125 of the revised manuscript).
2.2. Bioavailability of vitamin E
The bioavailability of vitamin E varies greatly between individuals and can be affected by several factors. The topic has been extensively reviewed in [21] and [22]; hence, it will be only briefly summarized here. Vitamin E in food, but especially in dietary supple-ments, is often present as an esterified molecule. After hydrolysis by pancreatic and intes-tinal digestive enzymes, vitamin E is released from food matrices and absorbed into mi-celles. Several proteins have been shown to affect the efficiency of vitamin E absorption in the intestine, namely the scavenger receptor class B type 1 (SR-BI), Niemann-Pick C1-like protein 1 (NPC1L1), and the ATP-binding cassette transporter of the sub-family A number 1 (ABCA1). The transport of vitamin E in blood is similar to that of cholesterol. Vitamin E in hepatocytes is transported and stabilized by α-TTP. As mentioned in Section 2.1., α-TOH is preferentially distributed to peripheral tissues due to its highest affinity to α-TTP. Thus, α-tocopheryl acetate, the stabilized form of α-TOH, and α-TOH are the most widely used forms of vitamin E in human trials. In addition to α-TTP, other cellular vitamin E binding proteins have been suggested in humans, such as supernatant protein factor (SPF) and tocopherol-associated protein (TAP). Possible factors that likely affect the bioavailabil-ity of vitamin E, i.e. its absorption and metabolism, such as the different forms of vitamin E itself, chemical modifications (e.g. esterification), the amount consumed at once, and in-teractions with other nutrients or drugs consumed in parallel, have been summarized by Borel et al. [22]. There are also host-related factors such as vitamin E status, state of health, and genetic factors. In fact, lower absorption of α-TOH in patients with metabolic syn-drome or the elderly has been reported [23,24]. It has been hypothesized that, probably due to increased inflammation and oxidative stress, the absorption in the small intestine as well as the hepatic trafficking of vitamin E are limited. These findings provide evidence that some populations may have a higher demand of vitamin E. From our point of, blood concentrations of vitamin E should be measured before and after supplementation in study participants to properly evaluate its effect in human trials, since the bioavailability of vitamin E, and in turn the vitamin E status, is affected by numerous factors and indi-vidual conditions. At present, the blood concentrations and the concentration in organs of α-TOH as well as the level of the endogenous metabolite of α-TOH, namely α-carboxyethyl-hydroxychroman (CEHC), are used as biomarkers of vitamin E status [25].
Reviewer 2
The manuscript by Liao et al. is carefully written, readable and the topic is of great interest. English quality is not bad but revision by an expert is necessary. I detected multiple small mistakes that will need fixing by an expert.
REPLY: We Thank the reviewer for the careful and valuable review of our manuscript. The revised manuscript has been finally proofread by an expert.
There are, however, some details i need to comment on, which will, in my opinion, improve the quality of the manuscript.
Major comments
- I think the article should start with the section 2., and not as it is in the original draft. The information comprehended in section 1. can be placed elsewhere and the readability of the text would improve. Moreover, it is the logical order to start talking first on vitamin E given the topic of the review.
REPLY: We are grateful to the reviewer for this suggestion. We have placed Section 1 first to briefly introduce the reader to the general concept of micronutrients playing crucial roles in the immune system before narrowing down to vitamin E (Section 2) which is the crux of the paper. For this reason, we have decided not to alter the order of Sections 1 and 2. We hope that the reviewer accepts our decision.
- In my opinion it is important to specify where vitamin E naturally occurs in dietary products. This information is missing.
REPLY: We thank the reviewer for bringing this to our attention. Sources of vitamin E have been added as suggested in Section 2 “Characteristics of vitamin E” (lines 63 to 65 of the revised manuscript) as follows.
“Vegetable oils such as wheat germ, sunflower, corn germ, soybean and rapeseed oil are the primary dietary source of vitamin E for humans. It is also found in some nuts, fruits and vegetables such as almonds, avocados, spinach and kale. [12]”
- Figure 1 should be placed closer to its first text mention. As it is, it is not optimal for the reader. In addition, the numbering in the legend is missing in the picture. Names in the figure and the legend should meet, e.g. ascorbic acid and vitamin C, etc.
REPLY: We thank the reviewer for the points raised. We mention Figure 1 in Section 2.4. for the first time. Accordingly, Figure 1 is placed directly following this notion. As suggested, we have deleted the numbering in the legend of Figure 1, since it is not present in the figure. We have also changed “ascorbic acid” in the legend of Figure 1 to “vitamin C” to fit with the figure, as suggested.
- The study mentioned in line 186-8 should be complemented with the date. It would be interesting to comment (if possible) if the result of this study are a trend or it has been like this for some time. May be also interesting to comment the situation in Europe/developed world and not limit it to Germany.
REPLY: We thank the reviewer for this interesting and valuable suggestion. We added the date at which the 2nd German Consumption Survey was done in line 222 of the revised manuscript. According to the online information, the 1st German Consumption Survey was conducted between 1985 and 1989. We agree with the reviewer that it would be interesting to compare the data and discuss the trend. Unfortunately, we cannot find any data of the 1st German Consumption Survey, nor in the online databank or our local library. We therefore kept the remaining section as is.
In lines 229 to 234 of the revised manuscript, we added the requested information about the intake of vitamin E in the USA, UK and Netherlands. The revised section reads as follows:
“In an assessment of several dietary surveys conducted in the USA, the UK, Germany and the Netherlands, Troesch et al. showed that more than 75 % of the population in the UK and USA consume amounts of vitamin E below the country-specific RDI. In the Netherlands, 25-50% of women and 5-25% of men were below the RDI [56].”
Minor comments
- Sentence in lines 27-28 is not very accurate: the term appearance has somehow mystic attributions. Authors should choose a more accurate term.
REPLY: We thank the reviewer for this suggestion. The word ‘appearance’ has been changed to ‘rise’ in line 27 of the revised manuscript.
- Intravascular coagulation in line 33 is redundant.
REPLY: We thank the reviewer for this correction. ‘Intravascular coagulation’ has been deleted in line 33 to address this comment.
- In line 48, it is more accurate to say: “the B vitamin group”; or some other formulation. Vitamin A (retinoids) have also immune system-related functions. This should be specified or beta-carotenes should be eliminated from the sentence.
REPLY: We thank the reviewer for this note, and we accordingly removed the misleading wording. Instead of “The B vitamins” the phrase now reads “the group of B vitamins” in line 48 of the revised manuscript.
- Sentences 303-4 are should be reformulated to have a more clear meaning. The name of the subsection 3.1.2. is not good and imprecise: beta-carotene is vitamin A too. Authors may want to revise the classification and nomenclature of vitamin A
REPLY: We thank the reviewer for this valuable suggestion. The revised statement now reads as follows: “The data also suggests that age, as well as the state of health of a person could impact the individual’s requirement of antioxidants”. Section 3.1.2 is now entitled as “vitamin A” instead of “vitamin A and carotene”.
- In sentence 305 more functions of vitamin A should be specified. It is incomplete
REPLY: We thank the reviewer for the careful proofreading of our manuscript and have expanded the part on the functions of vitamin A (lines 342 to 347 of the revised manuscript). The section reads as follows.
“Vitamin A is involved in various processes in our body: it is essential for (i) vision – vitamin A deficiency can lead to night blindness, (ii) growth and development – vitamin A is involved in the genetic regulation of cell and tissue formation, and (iii) immune function [16,81]. Furthermore, as described by Huang et al., (iv) vitamin A regulates the formation of epithelial tissues which act as the first barrier against pathogen invasion [16].”
- Table 1:
number of subjects in Bjelakovic study is wrong
The number of subjects in Tornwall and Kataja are exactly the same?
Subsections of the table should not be in accronyms (VC etc.) and again I do not agree with the denomination “VA and carotene”
The abbreviations should be corrected were mispelling and punctuation mistakes are and ordered alphabetically
REPLY:
We corrected the number of subjects for the study of Bjelakovic from 29,6707 to 296,707.
The study of Tornwall and Kataja were both carried out as part of the Alpha-Tocopherol, Beta-Carotene Cancer Prevention Study (ATBC). The study included a total of 29,133 participants. Tornwall focused on the effect on coronary events and Kataja-Tuomola on diabetes. We updated the number of subjects to the numbers including only in the study analysis of each study.
We changed the acronyms in the table to the full names and corrected “VA and carotene” to “vitamin A” only.
The paragraph “Because of the synergistic relationship…” was accordingly corrected (lines 378 to 385 in the revised manuscript) and punctuation mistakes in the legend were also corrected. The revised paragraph reads as follows:
“Due to the synergistic interaction between antioxidants with vitamin function, the supplementation with a mixture of vitamin E, β-carotene and vitamin C was tested in the prevention of oxidative stress-induced diseases including cancer, inflammation and metabolic disorders such as CVD and T2DM [105–109]. Previous reviews have summarized the results of several large-scale randomized controlled trials in which the effect of supplementations with multiple antioxidant vitamins was studied. They found that the risk of major cardiovascular outcomes is not significantly changed by the supplementation of multiple antioxidant vitamins including vitamin E and β-carotene [110,111].”
- For vitamin K there is also a non-specific and weak first sentence. Correction needed
REPLY: We thank the reviewer for this helpful notion. The statement (lines 397 to 398 of the revised manuscript) has been accordingly revised and reads now as follows:
“The term vitamin K comprises the molecules phylloquinone (vitamin K1) as well as the menaquinones (vitamin K2 or MKs).”
- When talking about the trace elements, it would be good to complete the info with the sources of these elements (e.g. where can Se be found in food etc.).
REPLY: We thank the reviewer for this important point. The requested information has been provided for each trace element (Mg: lines 482 to 484; Se: lines 552 to 554; Zn: lines 685 to 686 of the revised manuscript). The revised sections read as follows:
“Magnesium is the fourth most abundant mineral in the body, with the highest con-tent in bones followed by muscles, soft tissues and blood. Seeds, nuts and green leafy vegetables such as spinach are rich dietary sources of magnesium [132].”
“It is present most abundantly in some nuts (brazil nut), whole grains, dairy products, animal viscera and sea animal.”
“Main sources of zinc are seafood, cereals, turkey and lentils [177].”
Further remarks:
- Some spelling mistakes and a few other language corrections have been made, which are not highlighted in the revised manuscript.
- The “Track Changes” function of Microsoft Word conflicts with our bibliography software. For this reason, we highlighted all the revisions of the original manuscript in red color and struck through the words we want to delete.

Reviewer 2 Report
The manuscript by Liao et al. is carefully writen, readable and the topic is of great interest. English quality is not bad but revision by an expert is necessary. I detected multiple small mistakes that will need fixing by an expert.
There are, however, some details i need to comment on, which will, in my opinion, improve the quality of the manuscript.
Major comments:
I think the article should start with the section 2., and not as it is in the original draft. The information comprehended in section 1. can be placed elsewhere and the readability of the text would improve. Moreover, it is the logical order to start talking first on vitamin E given the topic of the review.
- In my opinion it is important to specify where vitamin E naturally occurs in dietary products. This information is missing.
- Figure 1 should be placed closer to its first text mention. As it is, it is not optimal for the reader. In addition, the numbering in the legend is missing in the picture. Names in the figure and the legend should meet, e.g. ascorbic acid and vitamin C, etc.
- The study mentioned in line 186-8 should be complemented with the date. It would be interesting to comment (if possible) if the result of this study are a trend or it has been like this for some time. May be also interesting to comment the situation in Europe/developed world and not limit it to Germany.
Minor comments:
- Sentence in lines 27-28 is not very accurate: the term appearance has somehow mystic attributions. Authors should choose a more accurate term.
- Intravascular coagulation in line 33 is redundant.
- In line 48, it is more accurate to say: “the B vitamin group”; or some other formulation. Vitamin A (retinoids) have also immune system-related functions. This should be specified or beta-carotenes should be eliminated from the sentence.
- Sentences 303-4 are should be reformulated to have a more clear meaning. The name of the subsection 3.1.2. is not good and imprecise: beta-carotene is vitamin A too. Authors may want to revise the classification and nomenclature of vitamin A
- In sentence 305 more functions of vitamin A should be specified. It is incomplete
- Table 1:
- number of subjects in Bjelakovic study is wrong
- The number of subjects in Tornwall and Kataja are exactly the same?
- Subsections of the table should not be in accronyms (VC etc.) and again I do not agree with the denomination “VA and carotene”
- The abbreviations should be corrected were mispelling and punctuation mistakes are and ordered alphabetically
- For vitamin K there is also a non-specific and weak first sentence. Correction needed
- When talking about the trace elements, it would be good to complete the info with the sources of these elements (e.g. where can Se be found in food etc.).
Author Response

(The authors gave the same response as above.)
